

# Impact of work environment perceptions and communication satisfaction on the intention to quit: an empirical analysis of nurses in Saudi Arabia

Abdulaziz M. Alsufyani[1,2], Khalid E. Almalki[3], Yasir M. Alsufyani[4], Sayer M. Aljuaid[2], Abeer M. Almutairi[5], Bandar O. Alsufyani[6], Abdullah S. Alshahrani[7], Omar G. Baker[1] and Ahmad Aboshaiqah[1]

[1] College of Nursing, King Saud University, Riyadh, Saudi Arabia
[2] Comprehensive Rehabilitation Center, Ministry of Human Resources and Social Development, Taif, Saudi Arabia
[3] Primary Health Care Center, Ministry of Health, Riyadh, Saudi Arabia
[4] College of Nursing, King Khaled University, Abha, Saudi Arabia
[5] King Fahad Specialist Hospital, Ministry of Health, Buridah, Saudi Arabia
[6] King Abdulaziz Specialist Hospital, Ministry of Health, Taif, Saudi Arabia
[7] Ahad Rufidah Hospital, Ministry of Health, Asser Region, Saudi Arabia

Corresponding author
Abdulaziz M. Alsufyani,
A.M.Alsofyani2@hrsd.gov.sa

## ABSTRACT

**Objective**. In consideration of the current nursing shortage in Saudi Arabia, we aimed to investigate the association among perceptions of work environment, communication satisfaction, and intentions to quit nursing profession among nurses. In addition, we aimed to investigate the mediating effect of communication satisfaction on the association between nurses' perception of work environment and their intentions to quit nursing profession.

**Methods**. This predictive correlational study was conducted at one of the major hospitals in Saudi Arabia from January 2020 to March 2020. It included a convenience sample of 367 full-time registered nurses who completed three types of close-ended questionnaires. We used IBM SPSS version 24.0 to analyze the collected data. Regression analyses were used to test the study's hypotheses. All regression assumptions were assessed and confirmed. Significance for all tests was set at $p \leq .05$.

**Results**. The findings indicated an affirmative association between work environment perception and communication satisfaction ($b = .764$, $p < .05$) among nurses. In addition, findings showed that work environment perception ($b = -.187$, $p < .05$) and communication satisfaction ($b = -.226$, $p < .05$) have negative impacts on the nurses' intentions to quit; indicating that as work environment perception or communication satisfaction increases, the intention to quit decreases among nurses. Further, a mediation effect of communication satisfaction on the relationship between work environment perception and intention to quit was confirmed.

**Conclusion**. This study presents a novel conceptual framework developed based on the literature about the predisposing factors for nurses' intentions to quit nursing profession. Our results suggest that work environment perception and communication satisfaction among the most contributing factors for nursesá¿£ resignation. Effective communication was established as a crucial factor for establishing attractive and healthy

working environment. Nursing managers can benefit by applying these findings to develop appropriate strategies to inhibit the shortage of nurses in Saudi Arabia.

## INTRODUCTION

In the Kingdom of Saudi Arabia, expenditure on healthcare keeps on rising. This is because health organizations are the central aspects of a healthcare system in Saudi Arabia as they significantly affect the health of the community. Nurses are among the largest and crucial workforces within health care organizations; they are accountable for guiding and providing optimum care. According to *Oliveira et al. (2017)*, concerns that adversely influence nurses elevate the probability of challenges in the delivery of optimal healthcare services. One of these challenges is nursing shortage which is dominant in Saudi Arabia (*Aboshaiqah, 2016*; *Alsufyani et al., 2020*).

However, it was established that nursing shortage was preceded by nurses' intentions to quit the profession. In addition, literature reports many factors linked to the nurses' resignation. For instance, *Nantsupawat et al. (2016)* stated that intention to quit nursing profession exacerbates by stressful work environment and poor communication climate and strategies. In addition, *Özer et al (2017)* established that nurses' satisfaction about work environment and communication climate within their healthcare organizations play an important role in their intentions to leave their jobs.

On the other hand, working environment constitutes a fundamental aspect for nurses' effective perception and competent delivery of care. In this context, work environment entails the social, physical, and psychological factors that influence and make-up the working conditions (*Raziq & Maulabakhsh, 2015*). In addition, work environment was defined by *Kohun (2005*, p. 27) as "the combination of efficiency factors, actions, and possible challenges associated with the performances and activities of individuals". Precisely, every element of this relationship is included in the organizational environment of individuals and their work setting (*Tsai, 2011*). The objective of building healthy and stable work environment is to enhance the quality of health services through retaining competent nurses and encouraging individuals' use of their skills, knowledge, and existing resources effectively and efficiently.

*Falatah & Conway (2019)* postulate that a poor workplace environment results in stress and dissatisfaction which increases employees' intentions to quit. Workplace factors such as working hours, administrative style, perceived workload, coworkers support, and sharing in decision-making fundamentally influence the decision of employees to remain or leave their nursing jobs (*Abdien, 2019*; *Masum et al., 2016*). In addition, *Admi et al. (2018)* pointed out that a poor work environment causes demoralization, dissatisfaction, and frustration. Further, problems in the working environment are worsened by a lack of communication, which, in turn, leads to greater intentions to quit.
Communication is a social-sentimental status that strengthens interpersonal relationships, collaborative teamwork, and interactions. Communication also leads satisfaction in the workplace; it encompasses the distribution of information with, and among personnel (*Hua & Omar, 2016*). Communication satisfaction is demonstrated as the level to which a person is satisfied with different elements of organizational interaction including work knowledge, personal feedback, interdepartmental communication, horizontal communication, communication with supervisors, the communication climate, and media quality (*Alshammari, Duff & Guilhermino, 2019*; *Andersson, 2018*). Poor communication results in information asymmetry and often, workplace conflicts which exacerbate misunderstanding between employers and employees. Poor communication in the nursing environment results in poor collaboration among team members in a unit, which adversely affects the quality of care offered to the patients. In addition, it deteriorates nurses' attachment to the organization, provoking their willingness to leave the nursing profession (*Doleman, Twigg & Bayes, 2020*).

The intention to quit the nursing profession among nurses in Saudi Arabia has gained more attention recently. Unfortunately, many of these studies showed massive and terrible rates of intentions of quitting among nurses. For instance, *Suliman (2009)* reported that the highest rate of intention to quit was 77.1% among bedside nurses, followed by 38% among nurse managers. *Al-Ahmadi (2014)* also reported high rate of turnover intentions among nurses in Saudi Arabia (60%). Surprisingly, the findings of *Al-Ahmadi (2014)* were congruent with a study conducted by *Saeed (1995)* 25 years ago in Riyadh, Saudi Arabia. More recently, a study conducted by *Alonazi & Omar (2013)* showed that the majority of the sample (75%) showed intentions to leave within 24 months. In this regard, *Al-Zayyer (2003)* estimated that the average length of expatriate nurses' stay in their jobs in Saudi Arabia was 43 months. In addition, Filipino nurses, followed by Indian nurses had high rate of intentions of quitting the nursing profession in Saudi Arabia (*Albougami et al., 2020*).

However, across the world, the impacts of working environment perception and communication satisfaction on nurses' intentions to quit have been documented. To our knowledge, although Saudi Arabia is currently experiencing shortage of nursing workforce and high rates of nurses with intentions to leave the profession, these variables have been scantly investigated. Therefore, this study intends to examine the association between nurses' perceptions of work environment, communication satisfaction, and their intentions to quit the nursing profession in King Faisal Medical Complex (KFMC), Saudi Arabia. In addition, this study aims to investigate the indirect effect of nurses' work environment perception on their intentions to quit through communication satisfaction. Consistent with the conceptual framework which guides this study, the following hypotheses were examined:

**H₁.** Nurses' perceptions of their work environment significantly predicts their intentions to quit nursing profession.

**H₂.** Nurses' perceptions of their work environment significantly predicts their satisfaction about communication in their institution.

**H₃.** Nurses' satisfaction about communication in their institution significantly predicts their intentions to quit nursing profession.

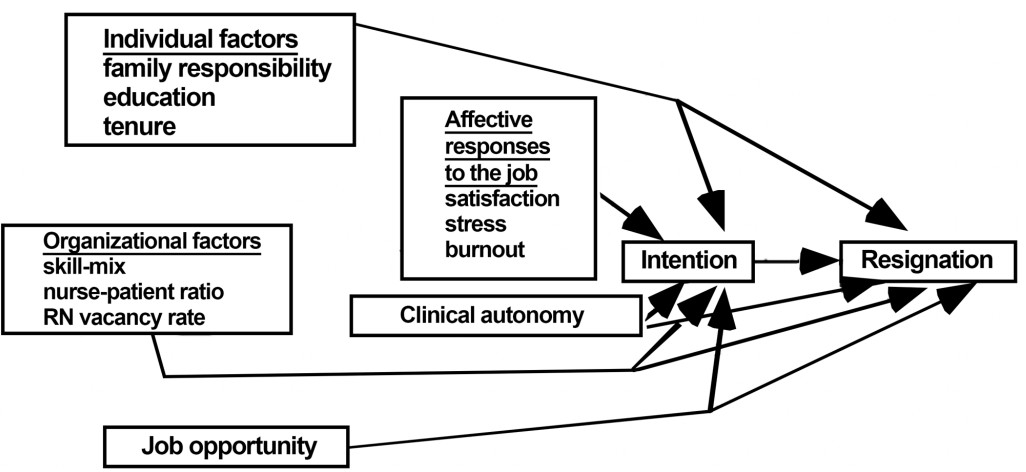

**Figure 1** Lake's model of nurse turnover (*Lake, 1998*).

**H₄**. Communication satisfaction acts as a mediator between nurses' perception of work environments and their intentions of quitting.

### Theoretical & conceptual framework

The model of nurse turnover proposed by *Lake (1998)* as shown in Fig. 1, was used to develop the study's conceptual framework. This model emphasizes nurses' intention to quit and personnel turnover as the main outcome variable. Job opportunity, individual factors, and work-related factors were included in the initial stage, while clinical autonomy and satisfaction were included in the middle phase of the affective responses. In the last stage of the framework, the actual turnover was resulted as an outcome of intention to quit the nursing profession.

We also added some new variables found in previous studies to modify Lake's nurse turnover model to discuss the relationship between intention to quit and nurses' work environment perception as shown in Fig. 2. This new model considers organizational factors as nursing work environment and affective responses to the job as communication satisfaction. This model is drawn based on the organizational factors and affective responses leading toward nurses' intentions to quit.

## MATERIALS & METHODS

### Research design and setting

This predictive correlational study was conducted at King Faisal Medical Complex in Taif, Saudi Arabia throughout the cross-sectional period from January 2020 to March 2020. This facility is known as one of the finest major hospitals in Saudi Arabia with a capacity of 800 beds.

### Sample and population

Nurses were recruited from King Faisal Medical Complex using a convenience sampling technique. Staff nurses included in this study were those with a minimum of two years of

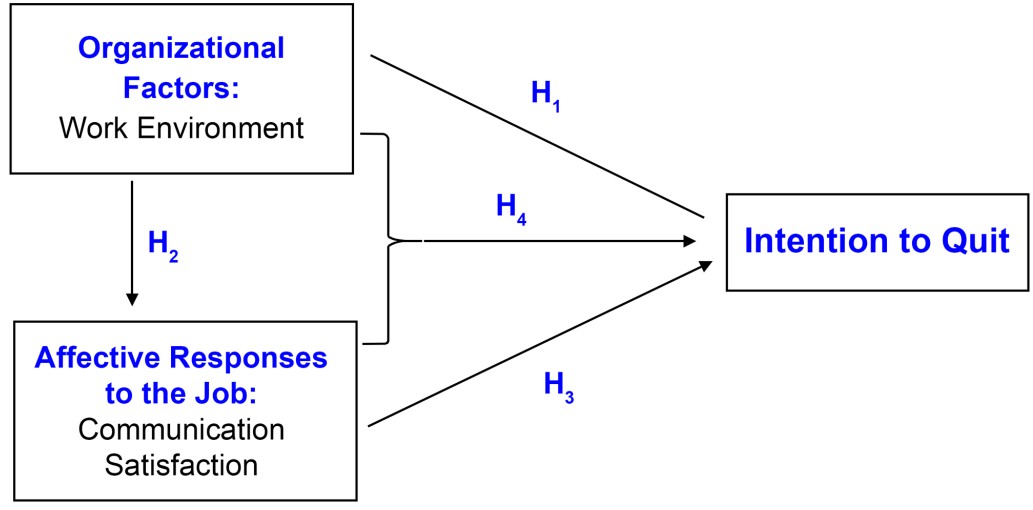

**Figure 2** The conceptual model for the current study.

experience designated to work at patients' bedsides. Nurses were included if they had a valid working license as a nurse by the Saudi Commission for Health Specialties and worked as full-time staff nurse. Exclusion criteria were paramedics, intern nurses, educators, managers, supervisors, and dependent nurses under supervision. In terms of sample size, a prior sample size estimation was calculated using 95% power, $\alpha = 0.05$, and medium effect size (0.15) with linear multiple regression: Fixed model, $R^2$ deviation from zero as the statistical basis of the calculation using G*Power 3.1.[TM]. As a result, the calculated sample size of 107 was deemed adequate to produce statistical differences.

## Research instruments

Three instruments were used in their original languages (English); including Practice Environment Scale of the Nursing Working Index (PES-NWI) (*Lake, 2002*); Communication Satisfaction Questionnaire (CSQ) (*Downs & Hazen, 1977*); and Intention to Quit Scale (*Cammann et al., 1979*). These instruments were covered with an appropriate socio-demographic questionnaire.

### Socio-demographic Questionnaire

Part of the survey used in this study included demographic profiles of the participants. Nurses' gender, age, nursing work experience, and nationality were assessed.

### Practice Environment Scale of the Nursing Work Index (PES-NWI)

The PES-NWI was used in the survey to examine the nurses' perception about the real work environment. The PES-NWI comprises five subdomains and a total of 31 items: nursing foundations for quality of care ($n = 10$), nurse participation in hospital affairs ($n = 9$), leadership and support of nurses ($n = 5$), staffing and resource adequacy ($n = 4$), and collegial nurse-physician relations ($n = 3$). A four-point Likert scale was used to measure the items of the PES-NWI, from *strongly disagree* = 1 to *strongly agree* = 4. Previously, *Almuhsen et al. (2017)* verified the validity and reliability of the scale, which vary from

0.87 to 0.96. In addition, this instrument was used in its original language. In this study, Cronbach's Alpha coefficient indicated the reliability of the PES-NIW as $\alpha = .94$.

### Communication Satisfaction Questionnaire (CSQ)

*Downs & Hazen (1977)* introduced a communication satisfaction questionnaire, including 25 questions from five sub-dimensions: organizational feedback, communication with a supervisor, communication climate, media quality, and organizational integration. A five-point Likert scale is used to measure the questionnaire items, from 1 = *strongly disagree* to 5 = *strongly agree*. Previously, *Greenbaum, Clampitt & Willihnganz (1988)* found internally reliable and consistent Cronbach's alpha scores across all organizations. In this study, Cronbach's coefficient alpha calculated for the reliability of the CSQ and found to be $\alpha = .91$.

### The intention to quit scale

*Cammann et al. (1979)* developed a scale for measuring employees' intentions to quit. The scale is scored using a five-point Likert scale, ranging from 1 = *strongly disagree* to 5 = *strongly agree*. This questionnaire has been used in previous studies and its inter-reliability has been established, varying from 0.83 to 0.92. In the present study, the reliability of this scale was found to be appropriate and valid with Cronbach's Alpha coefficient of $\alpha = .89$.

## Ethical approval

Ethical approval from an institutional review board was obtained from the Saudi Ministry of Health, ref. No. HAP-02-T-067. Permission to conduct this study was granted by the nursing management office of the King Faisal Medical Complex. The informed consents of the participants' agreement to participate in this study and their agreement to the researcher using their drawn data were gained. This study was consistent with the ethical principles of the Declaration of Helsinki. Declaration of Helsinki is a set of ethical principles concerning protecting the human beings in researches which was developed by the *World Medical Association (2013)*.

## Data collection procedure

Data were collected during a three-month period from January 2020 to March 2020. The participants completed the questionnaires and returned them to the designated place of submission after first receiving guidance about the study objectives and providing their written consent to participate. All participants were asked to submit their completed questionnaires within five days. Submissions were received in collection boxes allocated for this purpose at the KFMC.

## Data analysis

We used the IBM Statistical Package for Social Sciences (SPSS) version 24.0 to analyze the collected data. The nurses' demographic details were presented through descriptive statistics. As the data was normally distributed, the relationships between communication satisfaction, intention to quit, and work environment were determined using Pearson's Correlation coefficient ($r$). Regression assumptions (normality, homoscedasticity, linearity, independence of errors, and multicollinearity) were assessed to present valid results.

Normality was assessed using normal probability-probability (P-P) plot of residuals. As the assumptions of homoscedasticity and linearity relate to errors (*Field, 2018*), they were assessed by plotting the predicted values versus errors (*zpred* vs. *zresid*) on a scatterplot. Further, independence of errors was assessed using Durbin-Watson test for each model. Absence of multicollinearity was assessed through screening correlation matrix and Variance Inflation Factor (VIF).

The study's objectives were addressed using Baron and Kenny's (1986) proposed four-step approach to determine the presence of a mediation effect after confirming the presence of zero-order relationships among the study variables. This four-step approach requires conducting three simple linear regression (SLR) analyses as step 1, 2, and 3; to confirm the presence of zero-order relationships among the study's variables (*Baron & Kenny, 1986*). As significant relationships from step 1 through step 3 are confirmed, the researcher proceeds to step 4 which requires conducting multiple linear regression (MLR) analysis with force entry technique for the predictor and intervening variables predicting the outcome variable. If the results of model 4 show that the predictor variable is no longer significant when a mediator variable is controlled, the finding indicates a full mediation effect; if the predictor is still significant and a significant reduction in its relationship with the outcome variable has occurred, the findings supports a partial mediation effect (*Baron & Kenny, 1986*).

Simple linear regression analyses were conducted with the equation of $y = \alpha + bx$, where is "$y$" denoted for the predicted value of the dependent variable; "$x$" denoted for the independent variable; "$b$" is slope of the line; and "$a$" is $y$-intercept (*Gray, Grove & Sutherland, 2017*). In addition, the equation used for MLR analysis was as follows: $y = a + b_1 x_1 + b_2 x_2$ (*Shultz, Whitney & Zickar, 2014*); where is "$y$" denoted for the predicted value of the dependent variable; "$a$" denoted for $y$–intercept; "$b_1$" is the regression coefficient of the predictor variable ($x_1$); and "$b_2$" is the regression coefficient of the mediator variable ($x_2$). The acceptable probability value for all the statistical analyses was determined as $p \leq .05$.

# RESULTS

Of the 793 full-time nurses at the King Faisal Medical Complex, 500 received questionnaires and 367 completed surveys were returned, representing a response rate of 73.4%. Table 1 shows the demographic profile of the participants. The majority of the participants were female (78.7%), and their ages ranged from 31–40 years. Most nurses (56.1%) were in the "other" category of specialization, followed by OB/GYN, Med/Surg, and "critical," categories at 19.3%, 15.6%, and 9.0%, respectively. Additionally, nurses of many nationalities participated in this study, including Filipino (41.7%), Indian/Pakistani (25.3%), and Saudi (23.7%); "other nationalities" constituted 9.3% of the sample.

Table 2 shows that out of these three factors, the mean score for work environment (3.72 ± 0.63) was greater compared to the other two factors. The satisfaction of nurses regarding communication levels and work environment were moderate. Additionally, a moderate level of probability was reported for intentions to quit among nurses (2.47

**Table 1 Selected demographic characteristics of participants.**

| Variable | | Participants ($n = 367$) | |
|---|---|---|---|
| | | **F** | **%** |
| Gender | Male | 78 | 21.3% |
| | Female | 289 | 78.7% |
| Age groups | 25–30 years | 71 | 19.4% |
| | 31–40 years | 263 | 71.7% |
| | 41–50 years | 33 | 8.9% |
| Number of years worked | 2–10 Years | 103 | 28.1% |
| | 11–20 Years | 236 | 64.3% |
| | 21–30 Years | 28 | 7.6% |
| Nursing Area | Med/Surg | 57 | 15.6% |
| | OB/GYN | 71 | 19.3% |
| | Critical | 33 | 9.0% |
| | Other | 206 | 56.1% |
| Nationality | Saudi | 87 | 23.7% |
| | Filipino | 153 | 41.7% |
| | Indian / Pakistani | 93 | 25.3% |
| | Others | 34 | 9.3% |

**Table 2 Descriptive statistics and inter-correlations between work environment, communication satisfaction, and intention to quit.**

| Variables | Mean ± SD | 1 | 2 | 3 |
|---|---|---|---|---|
| Intention to Quit (1) | 2.47 ± 1.38 | (1) | | |
| Working environment (2) | 3.72 ± 0.63 | −.602-** | (1) | |
| Communication Satisfaction (3) | 3.71 ± 0.65 | −.581-** | .613** | (1) |

**Notes.**
** $p < .05$ (2-tailed).

± 1.38). A positive and significant relationship was revealed between work environment and communication satisfaction ($r = .613$; $p < .05$). These findings confirm the absence of multicollinearity between the predictors as their relationship was <0.8 and >0.3. In addition, the association between work environment perception and the intention to quit was found to be negative and significant ($r = −.702$; $p < .05$). Furthermore, the findings indicated that nurses' intentions to quit had inversely significant relationship with their communication satisfaction ($r = -.581$; $p < .05$).

The findings of regression assumptions tests showed no assumption was violated. Figure 3 shows that the residuals were normally distributed as they conformed to the fixed diagonal line of normality shown in the plot. In addition, the result of plotting the predicted values versus errors (*zpred vs. zresid*) on a scatterplot showed a random array of dots; indicating that the data met homoscedasticity and linearity assumptions of regression test. In addition, the absence of multicollinearity was confirmed by the result of VIF test which showed no variable has VIF greater than 10. Further, the results of Durbin-Watson
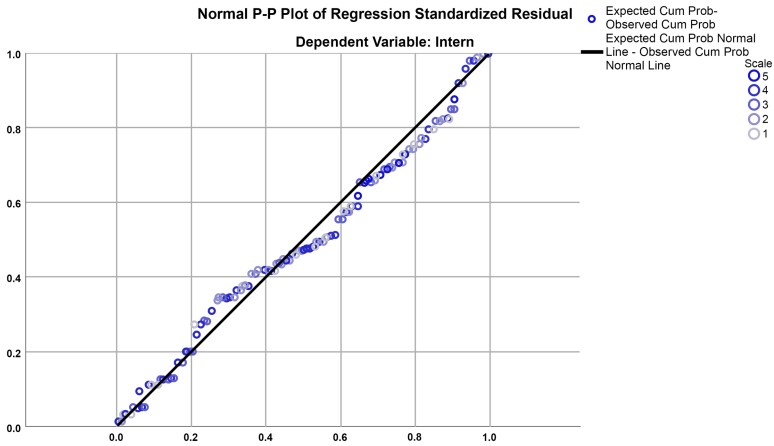

**Figure 3** Normal probability plot (P-P) of the regression standardized residuals.

**Table 3 Results of regression analyses—model summaries.**

| Model | Outcome variable | Predictors | b | Std. error | β | t | p | CI 95% |
|---|---|---|---|---|---|---|---|---|
| 1 | Intention to quit | (constant) | 29.053 | .597 | −.887 | 48.669 | <.05 | [27.879, 30.227] |
| | | Working environment | −.187 | .005 | | −36.772 | <.05 | [−.197, −.177] |
| | | $R = .89, R^2 = .79$ | | | | | | |
| | | $F = 1352.2; p < .05$ | | | | | | |
| | | Durbin Watson = 1.99 | | | | | | |
| 2 | Communication satisfaction | (constant) | 4.454 | 2.025 | .918 | .814 | .028 | [.471, 8.436] |
| | | Working environment | .764 | .017 | | 15.460 | <.05 | [.730, .798] |
| | | $R = .92, R^2 = .84$ | | | | | | |
| | | $F = 1950.8; p < .05$ | | | | | | |
| | | Durbin Watson = 1.97 | | | | | | |
| 3 | Intention to quit | (constant) | 28.311 | .573 | −.889 | 49.424 | <.05 | [27.184, 29.437] |
| | | Communication satisfaction | −.226 | .006 | | -37.046 | <.05 | [−.238, −.214] |
| | | $R = .88, R^2 = .79$ | | | | | | |
| | | $F = 1372.4; p < .05$ | | | | | | |
| | | Durbin Watson = 1.73 | | | | | | |
| 4 | Intention to quit | (constant) | 29.586 | .550 | −.455 | 53.804 | <.05 | [28.505, 30.668] |
| | | Work environment | −.096 | .012 | −.472 | -8.173 | <.05 | [−.119, −.073] |
| | | Communication satisfaction | −.120 | .014 | | −8.477 | <.05 | [−.147, −.092] |
| | | $R = .91, R^2 = .822$ | | | | | | |
| | | $F = 843.303; p < .05$ | | | | | | |
| | | Durbin Watson = 1.854 | | | | | | |

test confirmed the autonomy of residuals as its values varied from 1.73 to 1.99 in all models.

Table 3 presents that, in model 1, we regressed the nurses' perceptions of work environment against their intentions to quit the nursing profession using SLR analysis. The findings indicated a significant negative influence of working environment perception on the intention to quit among nurses ($b = −.187, p < .05$); indicating that as nurses' perception of work environment increases, their intentions to quit declines and vice versa. Further, it describes 79% of the total variation in nurses' intentions to quit ($F = 1352.2$;

$p < .05$). Therefore, it is confirmed that nurses' perception of work environment significantly predicts nurses' intention to quit, thus accepting $H_1$. In addition, based on the results of model 2, the formulated regression equation was as follows: *intention to quit* $= 29.053–0.187$ (*work environment perception*).

Table 3 also shows that, in model 2, we regressed the work environment perception of nurses against their communication satisfaction using SLR analysis. The findings present a statistically significant impact of work environment perception on the communication satisfaction among nurses ($b = .764$, $p < .05$); indicating that as work environment perception increases among nurses, their communication satisfaction increases proportionately. In addition, work environment explains 84% of the total variance for communication satisfaction ($F = 1950.8$; $p < .05$). Therefore, it is confirmed that nurses' perceptions of work environment significantly predict their satisfaction about communication inside their institution and, thus, $H_2$ was accepted. Based on that results, the formulated regression equation described as: *communication satisfaction* $= 4.454 + .764$ (*work environment perception*).

In addition, Table 3 shows that, in model 3, findings of SLR analysis indicated that nurses' communication satisfaction significantly predicts nurses' intentions to quit ($b = -.226$, $p < .05$) and thus $H_3$ was accepted. Further, nurses' communication satisfaction explains 78% of total variance in their intentions to quit nursing profession ($F = 1372.4$; $p < .05$). The formulated regression equation shown as: *intention to quit* $= 28.311- 0.226$ (*communication satisfaction*).

Lastly, as shown in Table 3, in model 4, when communication satisfaction was controlled, the relationship between nurses' work environment perceptions and their intentions to quit became considerably weak ($b = -.187$ to $b = -.096$, $p < .05$, 95% CI $[-.119, -.073]$); indicating a significant indirect effect of the nurses' perceptions of work environment on their intention to quit through their communication satisfaction as a mediator variable. In other words, communication satisfaction explains the association between the nurses' work environment perception and their intentions to quit; thus, accepting $H_4$. These results were presented into the MLR's equation as: *intention to quit* $= 29.586–0.096$ (*work environment perception*) $-0.120$ (*communication satisfaction*).

Figure 4 summarizes and presents the associations among the study variables in a schematic diagram.

## DISCUSSION

This study investigated the associations between nurses' work environment perception and their satisfaction about communication within the healthcare organization. Additionally, it investigated the impact of these variables on their intention to quit the nursing profession as these were found as determinants of intention to quit among nurses, which in turn, leads to resignation and staff shortage (*AbuAlRub et al., 2016*; *Albougami et al., 2020*; *Oliveira et al., 2017*; *Özer et al , 2017*).

The findings showed that the participants demonstrated a moderate level of positive perception regarding their work environment. These results are congruent with

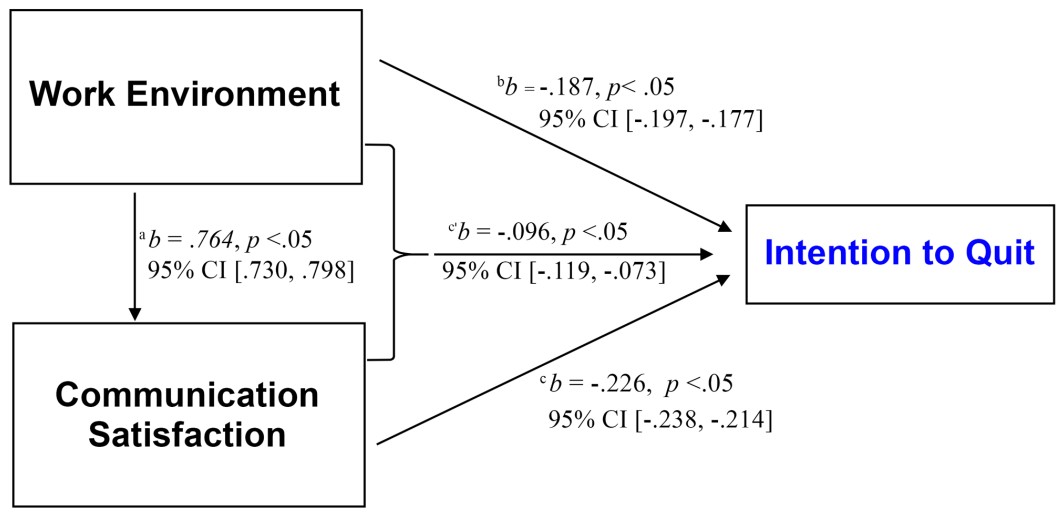

**Figure 4** **Summary of the relationships between study's variables.** Lowercase letter a indicates the association between work environment and communication satisfaction; b, the association between work environment and intention to quit; c, the association between communication satisfaction and intention to quit; c', the indirect effect of work environment on intention to quit through communication satisfaction as a mediator.

other studies conducted in Saudi Arabia (*AlMoosa et al., 2020*; *Almuhsen et al., 2017*), which showed moderate levels of satisfaction about work environment among multinational nurses. Concomitantly, international studies show comparable situations with no exceptions (*Al-Hamdan, Manojlovich & Tanima, 2017*; *Özer et al , 2017*; *Ulusoy & Polatkan, 2016*). However, these modest levels of work environment perception in Saudi Arabia are unexpected when they are compared to the total governmental expenses on the healthcare delivery system. In 2018, the governmental expenses on healthcare system reached 90 billion Saudi Riyals which constituted about 9.2% of total governmental budget (*Ministry of Health of Saudi Arabia, 2018*). However, it is reasonable to link these modest levels of work environment satisfaction with the current reported challenges in Saudi nursing practice. Per *Alsufyani et al. (2020)*, the status quo of nursing practice in Saudi Arabia is facing several challenges including shortage of staff, absence of clear and defined scope of practice, and lack of national and international benchmarking.

Likewise, the findings showed that the study's participants ranked their communication satisfaction within KFMC as moderate. Due to the dearth of local studies assessing communication satisfaction among nurses in Saudi Arabia, these findings were compared to international studies and are in line with previous studies (*Özer et al , 2017*; *Vermeir et al., 2018*). Although these studies investigated the importance of communication satisfaction and its role in enhancing job satisfaction, quality of care, and nurse retention, they showed moderate levels of communication satisfaction. In addition, these studies described and illustrated the presence of communication gap between top nursing management and bedside nurses. However, this communication gap indicates a presence of centralized management which restricts communication, sharing in decision-making, and professional

autonomy. Further, centralized management was also reported by *Alsufyani et al. (2020)* as one of the challenges in nursing practice in Saudi Arabia. In addition, almost all of healthcare organizations in Saudi Arabia depend heavily on non-Saudi nurses from various nationalities with different native languages such as: English, Hindi, Malayic, Tagalog, Tamil, and Urdu. Hence, it is important to foster communication and feedback within Saudi Arabian healthcare organizations through training programs concerning communication skills.

The findings of this study revealed a medium-level of intention to quit among nurses. These findings are consistent with other Saudi Arabian studies (*Albougami et al., 2020*; *Alsaqri, 2014*) and other overseas studies (*Bal, 2013*; *Danayiyen, 2015*; *Özer et al , 2017*). Based on the situation in Saudi Arabian healthcare organization, which relies heavily on expatriate nurses to deliver care, the findings of this study must be taken seriously to develop and execute systematic strategic plans that are aimed at achieving an attractive nursing work environment, to minimize these confounding rates of intentions to leave the profession among nurses.

On the other hand, the findings showed an affirmative positive association between work environment perception and nurses' satisfaction about communication within healthcare organization. Further, it explained 84% of the total variance in communication satisfaction among nurses. It was established that if the perception of work environment improves, the communication satisfaction improves proportionately. These findings are congruent with Turkish study conducted by *Özer et al (2017)*. Undoubtedly, communication satisfaction is a key ingredient in creating healthy work environment and vice versa. Therefore, it is reasonable to assume that healthy work environment is a cornerstone for establishing skilled communicators.

The study's findings offer valuable insight into the link between the work environment perception and the nurses' intentions to quit the nursing profession. The results of this study established that nurses' satisfaction level with their work environment was inversely proportional to their intentions to quit. Specifically, higher the nurses' satisfaction with work environments, the lower the intentions to quit. These findings are consistent with previously conducted studies by *Al-Hamdan, Manojlovich & Tanima (2017)*, *Özer et al (2017)* and *Zhang et al. (2014)*, in Jordan, Turkey, and China, respectively, whereby nurses have linked their intentions to quit mainly to their work environment conditions.

Similarly, a negative association has been detected between nurses' satisfaction about communication within their healthcare organization and their intentions of turnover. In other words, it indicated that improving communication strategies could inhibit the intentions to quit nursing profession among nurses. These findings support the assertion of *AlMoosa et al. (2020)*, that fair treatment and satisfied communication climate minimize the nursing shortage. Further, these findings are congruent with the findings of overseas studies; the same association was detected in studies conducted by *Özer et al (2017)* and *Vermeir et al. (2018)*, in Belgium and Turkey. In addition, (Mustamil et al., 2014), concluded that nurses' satisfaction about communication is a crucial factor in retaining nurses.

Lastly, the mediation effect of communication satisfaction on the association between work environment and intention to quit among nurses was revealed by the findings of this study. These findings affirmed that communication satisfaction explained the relationship between work environment perception and intention to quit among nurses. These findings are supported by the proposed standards of American Association Critical-Care Nurses (AACN) to create healthy nursing work environment (*American Association of Critical-Care Nurses, 2005*). These standards portray skilled communication as a crucial factor for establishing an attractive and healthy work environment, which, in turn, leads to minimizing nurses' intention to quit the profession. In addition, similar findings were reported by Turkish study conducted by *Özer et al (2017)*. However, these findings illustrate the importance of effective communication climate on nurses' outcomes.

In term of study's limitations, this study was conducted in a specific healthcare organization, King Faisal Medical Complex in Saudi Arabia, with conveniently selected sample, which may minimize the representation of the population. In addition, this study focused on only two factors that were found to inhibit the intention to quit the profession while ignoring other equally important factors such as leadership style, professional promotion, and workplace violation rates. Further, our data in this study was collected through self-administered tools which augment potentiality for bias as the researchers were unable to ensure that nurses reported their own perceptions honestly. Thus, there is a need for future studies with mixed methods or qualitative designs to ensure a wider range of understanding of the phenomenon of quitting the profession among nurses in Saudi Arabia.

However, in terms of implications for nursing management, the findings of this study present a novel theoretical framework based on the previous knowledge and literature about the factors that affect nurses' intentions to quit nursing profession and to leave their organizations. The study's findings offer valuable evidence that a healthy work environment and effective communication strategies within a healthcare organization positively influence nurses' retention. These results lead to our recommendation that the shortage of nurses in Saudi Arabia can be managed by appropriate nursing management that enhances communication satisfaction and work environments, as these improvements would reduce nurses' intention to quit. In addition, we emphasize the need for establishing effective communication strategies and channels with periodic assessments to ensure their effectiveness.

## CONCLUSION

This study highlighted two important factors affecting the nurses' intentions to quit nursing profession in Saudi Arabia. Our results suggest that modest levels of work environment perception and communication satisfaction among nurses were seen as indicators for further systematic strategic plans to achieve attractive nursing working environment. These factors were found to contribute and predict nurses' intention to leave and resign. In addition, our findings suggest that communication climate was established as a crucial factor in establishing attractive and healthy working environment. However, this study

presents a novel conceptual framework that has been developed based on the previous knowledge and literature for the contributing factors for nurses' intention to quit nursing professional and shortage nurses.

## ACKNOWLEDGEMENTS

The authors are very thankful to the Deanship of Scientific Research at King Saud University, Saudi Arabia (Research Group No. RG-1436017). In addition, all the associated personnel in any reference that contributed in/for the purpose of this research are also acknowledged.

### Funding

This study was funded and supported by the Deanship of Scientific Research at King Saud University, Saudi Arabia through Research Group No. RG-1436017. The funders had no role in study design, data collection and analysis, decision to publish, or preparation of the manuscript.

### Grant Disclosures

The following grant information was disclosed by the authors:
Deanship of Scientific Research at King Saud University, Saudi Arabia through Research Group: RG-1436017.

### Competing Interests

The authors declare there are no competing interests.

### Author Contributions

- Abdulaziz M. Alsufyani conceived and designed the experiments, performed the experiments, analyzed the data, authored or reviewed drafts of the paper, and approved the final draft.
- Khalid E. Almalki conceived and designed the experiments, analyzed the data, authored or reviewed drafts of the paper, and approved the final draft.
- Yasir M. Alsufyani performed the experiments, analyzed the data, authored or reviewed drafts of the paper, and approved the final draft.
- Sayer M. Aljuaid and Abdullah S. Alshahrani conceived and designed the experiments, performed the experiments, prepared figures and/or tables, authored or reviewed drafts of the paper, and approved the final draft.
- Abeer M. Almutairi analyzed the data, prepared figures and/or tables, authored or reviewed drafts of the paper, and approved the final draft.
- Bandar O. Alsufyani conceived and designed the experiments, prepared figures and/or tables, and approved the final draft.
- Omar G. Baker conceived and designed the experiments, performed the experiments, analyzed the data, prepared figures and/or tables, and approved the final draft.

- Ahmad Aboshaiqah conceived and designed the experiments, performed the experiments, analyzed the data, prepared figures and/or tables, authored or reviewed drafts of the paper, and approved the final draft.

## Human Ethics

The following information was supplied relating to ethical approvals (i.e., approving body and any reference numbers):

The Institutional Review Board of Ministry of Health in Saudi Arabia approved this research (No. HAP-02-T-067).

## Data Availability

Raw data are available as a Supplementary File.

## Supplemental Information

Supplemental information for this article can be found online at http://dx.doi.org/10.7717/peerj.10949#supplemental-information.

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
