# Peer review of "Impact of work environment perceptions and communication satisfaction on the intention to quit: an empirical analysis of nurses in Saudi Arabia"

_PeerJ, doi:10.7717/peerj.10949_

## Round 0.1 · original submission · Major Revisions

Your manuscript has been reviewed and requires several modifications prior to making a decision. The comments of the reviewers are included at the bottom of this letter. Reviewers indicated that methods and results sections should be improved. The literature review does not indicate the significance of the study. Why this study was very important to address? What was the gap in knowledge? The manuscript also needs biostatistical consulting because there are several statistical errors. I agree with the evaluation and I would, therefore, request for the manuscript to be revised accordingly. In addition to these; please see my comments below:

In Abstract: Change “SPSS” to “IBM SPSS Version 24.0”.

In Manuscript: Change “P<0.00” to “P<0.05”.

In Line 181: Correct the research design “descriptive, correlational study” with appropriate design.

In Line 192-193: Which statistical package or tool was used to calculate sample size? Please provide the name of the package or the tool.

In Line 195-198: These scales were tested for reliability and validity in the English language version, but survey instruments must be retested when translated to other languages. Each time a scale is translated into a new language, the reliability and validity should be retested in that language to ensure that the survey is still fulfilling its purpose in the new language. Did a group of experts check the validity of the Arabic version of the scales?

In Line 246: The methods section needs more elaboration, in particular, data analysis. For example, did you perform a step-wise/forward/backward regression? How did you check the normality assumption of the data? Please check your methods section with an expert biostatistician.

In Line 251: It is not clear for me why the authors prefer to use the PROCESS tool rather than IBM SPSS?

In Line 254: Is there a typo for “p < 0.005”? If yes, please correct the p-value.

In Table 1: Please recalculate the percentages of each variable, because the total percentage is less than 100% in some variables (for example, gender total 99.9%). Use 1 digit for decimals of total percentages being consistent with other groups’ percentages.

In Table 2: If you use a nonparametric test, you should give median and interquartile range values rather than mean and standard deviation values.

Reviewer 1 ·

Basic reporting

This is interesting study which covered very important topic that still in hot discussion in Saudi nursing community and healthcare in general. The paper were well written, however, there some parts need revisions for clarity and flow of writing. The manuscript were very well structured and figures were clear. The results were connected to the research objectives.

Experimental design

The study have been investigated very important issues in Saudi Arabia. The research hypotheses were well stated. There was clear explanation to the ethical approval and how the data was collected. Methods were explained clearly, however, some areas need justifications.

Validity of the findings

Statistical tests need to be justified/or revised.

Additional comments

This is very interesting study which discussed the attrition of nursing workforce in Saudi Arabia. I have made some comments and hopefully they benefit the enhancement of the manuscript:
In the abstract:
''In addition, we aimed to explore the mediator effect of communication satisfaction on the association..''
I think ''explore'' is not the right word here maybe investigate or examine.
Please write the SPSS version in the abstract as well.

Introduction:
Very well written, defining all the concepts that related to the current research.
''It is a behavioral tendency—or the cognitive expression associated to the behavioral decision—of an individual toward quitting his or her employment. Specifically, employees᾿ intentions to quit often arises because of the physical end results of task allocations within the firm.'' please provide reference here.

''Long working hours comprise one of the challenges encountered by personnel within in the Saudi health system. Additionally, the lack of appropriate training opportunities for nurses to advance their careers and of opportunities to work independently is another important challenge. Depressive and complicated work circumstances are further causes that lead nurses to leave their existing jobs in health institutions in Saudi Arabia.'' they maybe very common facts about the current situation in Saudi Arabia related to the nursing workforce, but please add references to support your claims.

Literature review:
The literature review needs revision to enhance the flow of the writing. Please revise each paragraph and ensure it is consistent with its topic sentence. Some of the paragraphs have general topic sentences which are not relevant to the entire paragraphs.

It will be good adding Saudi studies that covered the current topic for example related to work satisfaction in Saudi Arabia. Same apply to work environment and attrition of nurses in Saudi Arabia. Was there any recent statistic in Saudi about the current attrition rate?

The literature review does not indicate the significance of the study. Why this study was very important to address? What was the gap in knowledge?

Methods:
Given that it was not clear from the literature review why this study was conducted using descriptive, correlational design, it would be good adding a sentence in the research design justifying the choice of the current used design.
I expect this to be reinforced as last paragraph in the review section.
sample and population: Valid and active, were some of the nurses who work in that hospital either part time or full time nurses did not have valid and active licenses? Please revise this sentence.
Please justify why 40 hours per week and why two years of experience. Is there a reason relevant to the current research?
It was good mentioning each used scale with information related to its validation. Just one minor comment, please reorder the information about them in the manuscript according to how they appeared in the study’s questionnaire and under the section about research instruments.
What is Helsinki declaration in the ethical approval? please explain it for the readers.
It was not clear why there was mixing between parametric and non parametric statistical tests. If the data were not normally distributed Spearman’s correlation test will be the good choice as you did to investigate the relationship between the variables. However, median and IQR will be the best to present these data instead of mean and SD.
Can you please explain PROCESS tool in the data analysis?

Results:

Revise the written numbers to match the tables. The majority of the participants were female 7(8.7%), please revise this written sentence.
Discussion and conclusion:
As the literature review please revise the flow of the writing under the discussion section.
It will be great connecting your findings to what was already found locally (in Saudi Arabia) and internationally. Then you can build on this knowledge by writing about what your study added to this knowledge. From the current writing it is not clear the strength of your study's results and what their contribution to the knowledge.

Not clear how these conclusion and recommendations were drawn from. I suggested revising the discussion section to enable the readers to demonstrate the link between your discussion, recommendations and conclusion. Please revise the recommendations and conclusion part for clarity and maintain their alignment with the discussion of the results. As well. please revise the conclusion part in the abstract to make it stronger. The conclusion part looks very general and does not demonstrate the importance of the study's results.

Overall, it is really good study and will benefit the policy makers in Saudi Arabia to consider the factors that affect the attrition rate of nurses as stated in the manuscript. However, it needs revisions to demonstrate the strength of this study and clarify its addition to the current knowledge.

Reviewer 2 ·

Basic reporting

The study analyzed the risk factors for intention to quit and found that improving communication satisfaction and work environments for nurses could reduce their intentions to quit. However, this conclusion is too strong that cannot be supported by current study design. this should be rephrased as an observational study can at best provide preliminary results. This conclusion can only be supported by RCT (different levels of communication satisfaction). for most of the times, modifying risk factor may not help to change the outcome.

Experimental design

1. The conclusion cannot be supported by the study design.
2. The authors proposed a novel framework in figure 2. But I think this can be better represented in a structural equation modeling framework (Zhang Z. Structural equation modeling in the context of clinical research. Ann Transl Med. 2017;5(5):102. doi:10.21037/atm.2016.09.25); they currently employed logistic regression model to address this model, which is inappropriate. This should be discussed and data be reanalyzed to see robustness of the current finding.
3. Within the causal inference framework, the authors can establish the acyclic directed graph; but current analysis is not methodologically correct. for instance, work environment is a confounder based on figure 3; so the independent effect of work environment on intention to quit should be -0.226; however, there should be many other confounders , which the authors did not fully accounted for. thus, the independent effect of -0.226 is not valid.

Validity of the findings

1. The current finding cannot be supported by the analysis/study design.

Additional comments

1. The abstract cannot expess the main idea of the study; especially, the result section is poorly written without any useful information; you need data to support your conclusion in the abstract to make it stand alone.
2. intention to quit can be influenced by numerous other factors such as financial issues, work load and so on. How can you address these factors?

Reviewer 3 ·

Basic reporting

For a research article, there is no need for a separate section for "Literature Review" (page 9). Please incorporate this as part of the Introduction section.

Please distinguish clearly between theoretical framework and conceptual framework. You mentioned theoretical framework on page 8, along with Figure 1 and Figure 2. However, your Figure 2 appears to be the conceptual framework of the study.

Table 3 is confusing. You reported results of regression models. I can only see independent variable in the model(s). Where is the dependent or criterion variable?
Why did you conduct MLR with only 1 independent variable and 1 dependent variable?
What is the regression equation?
In MLR, researcher uses several independent/predictor variables to predict the outcome of a dependent/criterion variable. This is not reflected in this study.

Experimental design

The Methods section needs more elaboration, in particular, data analysis.

It is inadequate to just mention multiple regression analysis. For example, you must clearly identify the independent/predictor variables (IV) as well as the dependent/criterion variables (DV) used in the MLR. Did you conduct a step-wise/forward/backwards regression? etc.

I am also unclear why you brought in Spearman's Rank correlation. You can get a correlation matrix as part of the findings when you conduct MLR.
Furthermore, to conduct MLR, both the IV and DV must be continuous variables. Spearman's Rank correlation is for ordinal data.
Looking at your hypotheses and results, I would think you wanted to do a prediction study. Please make sure that the statistical tests used is correct and clearly described in data analysis.

Validity of the findings

The authors used three study instruments in this study, However, reliability and validity of the findings were not reported.

Additional comments

1. Please improve on the Abstract. It does not accurately reflect the content of the manuscript, in particular, Results.
2. There is a need to distinguish between between theoretical framework and conceptual framework and use them appropriately.
3. Methodology of the study needs more elaboration, in particular, data analysis.
4. Results of the study need to be clearly reported. Please recheck Table 3.

Annotated reviews are not available for download in order to protect the identity of reviewers who chose to remain anonymous.

---

## Round 0.2 · Major Revisions

The manuscript has been assessed by three reviewers. I agree with Reviewer 3 agree that there are still a few points that need to be addressed, especially in the statistical methods used in data analysis and hence the results. We would be glad to consider a substantial revision of your work, where the reviewer’s comments will be carefully addressed one by one. In addition to these;

-In Line 489: Correct “An a prior sample size” to “In a priori power analysis, the sample size estimation was calculated as…”.

-Compress tables 3-4-5-6 as a one-table file.

-Why did researchers use the F test for power analysis? It was not mentioned that the nurses were grouped. How many groups were there and what were the groups? While reaching this number in the calculation made for power analysis, which status was chosen? "fixed effects, omnibus, one-way" or "fixed effects, special, main effects and interactions". I could not reach the number you calculated for different group numbers. Please check the power analysis calculation details and share them with me.

Reviewer 1 ·

Basic reporting

Thank you to the authors for addressing my feedback as (reviewer 1). Thanks for editing the paper. I have reviewed the new changes, and I am happy with the paper in the latest version to be accepted. I still think this study is valuable and worth publication. Hopefully, the readers will consider the findings to help nurses to continue in the profession.

Experimental design

no comment

Validity of the findings

no comment

Additional comments

no comment

Reviewer 2 ·

Basic reporting

My previous comments have been well addressed.

Experimental design

good

Validity of the findings

good

Additional comments

good

Reviewer 3 ·

Basic reporting

There were numerous grammatical errors in the manuscript. Editing to correct the errors is necessary.

Sufficient background of the study was provided.

There were six tables and four figures - far too many for a manuscript. Please consider to reduce the number of tables. Note: In the previous manuscript, there were three tables but it has increased to six in this revised manuscript.

Experimental design

In the previous manuscript, the dependent variable(s) was/were not identified. The author used MLR as the statistical method to analyse the data.
However, in the revised manuscript, there were two independent / predictor variables (work environment, communication satisfaction) and two dependent / criterion variables (communication satisfaction, intention to quit). I also noticed that you were using communication satisfaction as both an independent variable (see Table 5 & Table 6) and a dependent variable (see Table 3).
Please note these two points:
(i) MLR is used for predicting the value of one dependent variable from the values of two or more independent variables
(ii) a variable cannot be both an independent variable and a dependent variable at the same time

Furthermore, in the revised manuscript, you were conducting some simple linear regressions, where there was only one independent variable and one dependent variable (see Tables 3, 4 & 5).

Validity of the findings

When the statistical methods used were questionable, validity of the findings would be affected.

Additional comments

I find this revised manuscript just as confusing, in particular data analysis and results. Please relook at the statistical methods used to anlalyse your data. Be very careful when MLR could be used.

Reduce the number of tables used.

The manuscript needs English editing.

Annotated reviews are not available for download in order to protect the identity of reviewers who chose to remain anonymous.

---

## Author Rebuttal · Round 0.2

**[October-30-2020]**

*Journal of Life and Environmental Science*

## Dear Editors:

We thank the reviewers for their generous comments on our manuscript. We would like to inform you that we have edited the whole manuscript to address their comments accordingly.

We believe that the manuscript is now suitable for publication in your valuable Journal of Life and Environmental Science.

**Abdulaziz M. Alsufyani**

**College of nursing, King Saud University, Saudi Arabia**

*Abdulaziz Alsufyani*

**On behalf of all co-authors**

# Editor comments (Aslı Suner):

- In Abstract: Change "SPSS" to "IBM SPSS Version 24.0".

  *Agreed, we have changed "SPSS" to "IBM SPSS Version 24.0" in Line 32.*

- In Manuscript: Change "$p < 0.00$" to "$p < 0.05$".

  *Agreed. we have changed "$p < 0.00$" to "$p < 0.05$" in whole manuscript.*

- In Line 181: Correct the research design "descriptive, correlational study" with appropriate design.

  *Agreed. we have changed the research design to the appropriate design in line 29 and line 149.*

- In Line 192-193: Which statistical package or tool was used to calculate sample size? Please provide the name of the package or the tool.

  *Agreed. we have provided the required information in Line 159-162.*

- In Line 195-198: These scales were tested for reliability and validity in the English language version, but survey instruments must be retested when translated to other languages. Each time a scale is translated into a new language, the reliability and validity should be retested in that language to ensure that the survey is still fulfilling its purpose in the new language. Did a group of experts check the validity of the Arabic version of the scales?

  *Dear editor, thanks for your generous comments. Actually, these scales were used in their original languages (English) as they were shown in literature as valid and reliable scales. We have presented more clarification in Line 165.*

- In Line 246: The methods section needs more elaboration, in particular, data analysis. For example, did you perform a step-wise/forward/backward regression? How did you check the normality assumption of the data? Please check your methods section with an expert biostatistician.

  *Agreed, in cooperation with an expert biostatistician, we have provided the required information regarding regression assumptions and models techniques in Line 222-235. In addition, all assumptions were assessed and confirmed by results in line 263-270.*

- In Line 251: It is not clear for me why the authors prefer to use the PROCESS tool rather than IBM SPSS?

  *Dear editor, thanks for your comments. Although the process tool gives the same results of MLR as shown in (Hayes, Andrew F., et al. "The Analysis of Mechanisms and Their Contingencies: PROCESS versus Structural Equation Modeling." Australasian Marketing Journal (AMJ), vol. 25,1, 2017, pp. 76–81., doi:10.1016/j.ausmj.2017.02.001), we have changed it and used IBM SPSS accordingly.*

- In Line 254: Is there a typo for "$p < 0.005$"? If yes, please correct the p-value.

  *Agreed. we have corrected the typo in Line 240.*

- In Table 1: Please recalculate the percentages of each variable, because the total percentage is less than 100% in some variables (for example, gender total 99.9%). Use 1 digit for decimals of total percentages being consistent with other groups' percentages.

  *Agreed. we have corrected all percentage in Table 1.*

- In Table 2: If you use a nonparametric test, you should give median and interquartile range values rather than mean and standard deviation values.

  *Dear editor, thanks for your comment, we have used parametric test (Pearson's Correlation test) but it was a mistake in writing; we wrote Spearman instead of Pearson. However, it is corrected in Line 221.*
* * *
# Reviewer 1 comments (Anonymous):

## In the abstract:

- "In addition, we aimed to explore the mediator effect of communication satisfaction on the association..." I think "explore" is not the right word here maybe investigate or examine.

  *Agreed. we have changed the word in Line 26.*

- Please write the SPSS version in the abstract as well.

  *Agreed. we have changed the word in Line 32.*

## Introduction:

- Very well written, defining all the concepts that related to the current research.

  *Appreciated. THANKS.*

- "It is a behavioral tendency—or the cognitive expression associated to the behavioral decision—of an individual toward quitting his or her employment. Specifically, employees' intentions to quit often arises because of the physical end results of task allocations within the firm." please provide reference here.

  *Dear editor, based on an external editor opinion, this sentence was removed and replaced by an appropriated sentence.*

- "Long working hours comprise one of the challenges encountered by personnel within in the Saudi health system". Please provide reference here.

  *Dear editor, based on an external editor opinion, this sentence was removed and replaced by an appropriated sentence.*

- "Additionally, the lack of appropriate training opportunities for nurses to advance their careers and of opportunities to work independently are other important challenges". Please provide reference here.

  *Dear editor, based on an external editor opinion, this sentence was removed and replaced by an appropriated sentence.*

## Literature review:

- The literature review needs revision to enhance the flow of the writing. Please revise each paragraph and ensure it is consistent with its topic sentence. Some of the paragraphs have general topic sentences which are not relevant to the entire paragraphs.

  *Agreed, we have reviewed it and presented it in logical and consistent forms.*

- It will be good adding Saudi studies that covered the current topic for example related to satisfaction in Saudi Arabia. Same apply to work environment and attrition of nurses in Saudi Arabia. Was there any recent statistic in Saudi about the current attrition rate?

  *Agreed, we have reviewed them and presented it in the background in Line 101-112.*

- The literature review does not indicate the significance of the study. Why this study was very important to address? What was the gap in knowledge?

  *Agreed, we have reviewed it and presented the required information in Line 113-117.*

## Methods:

- Given that it was not clear from the literature review why this study was conducted using descriptive, correlational design, it would be good adding a sentence in the research design justifying the choice of the current used design.

  *Agreed, dear editor, the study's design was corrected appropriately in Line 29 and Line 149.*

- Sample and population: Valid and active, were some of the nurses who work in that hospital either part time or full time nurses did not have valid and active licenses? Please revise this sentence.

  *Dear editor, of course, in that hospital there were newly hired expatriate nurses who need at least 4 months to complete the required requirements for applying Saudi Nursing Licensing Exam (SNLE) and during that time they were working as nurses based on their nursing license from their mother country. So, we meant valid and active Saudi Nursing License. In addition, this sentence was revised in Line 156-158.*

- It was good mentioning each used scale with information related to its validation. Just one minor comment, please reorder the information about them in the manuscript according to how they appeared in the study's questionnaire and under the section about research instruments.

   *Agreed, we have reordered the information in Line 165-169.*

- What is Helsinki declaration in the ethical approval? please explain it for the readers.

   *Agreed, we have provided the required information in Line 204-205.*

- It was not clear why there was mixing between parametric and non-parametric statistical tests. If the data were not normally distributed Spearman's correlation test will be the good choice as you did to investigate the relationship between the variables. However, median and IQR will be the best to present these data instead of mean and SD.

   *Agreed, we have assessed and confirmed the regression assumptions and used Pearson's correlation as well. In addition, all information was corrected in Line 263-270.*

- Can you please explain PROCESS tool in the data analysis?

   *Dear editor, this tool was removed and MLR used accordingly.*

**Results:**

- Revise the written numbers to match the tables. The majority of the participants were female 7(8.7%), please revise this written sentence

   *Agreed, we have provided the information in Line 245*

**Discussion and conclusion:**

- As the literature review please revise the flow of the writing under the discussion section.

   *Agreed, we have revised it and presented it in consistent form in Line 314-402.*

- Please revise the recommendations and conclusion part for clarity and maintain their alignment with the discussion of the results. As well.

   *Agreed, we have revised it and presented it in consistent form in Line 403-412.*

- Please revise the conclusion part in the abstract to make it stronger. The conclusion part looks very general and does not demonstrate the importance of the study's results.

   *Agreed, we have revised it and presented it in consistent and strong, see Abstract.*
* * *
# Reviewer 2 (Anonymous)

**Experimental design**

▪ The conclusion cannot be supported by the study design.

*Agreed, the study's design was corrected appropriately in Line 29 and Line 149.*

▪ The authors proposed a novel framework in figure 2. But I think this can be better represented in a structural equation modeling framework (Zhang Z. Structural equation modeling in the context of clinical research. Ann Transl Med. 2017;5(5):102. doi:10.21037/atm.2016.09.25); they currently employed logistic regression model to address this model, which is inappropriate. This should be discussed and data be reanalyzed to see robustness of the current finding.

*Dear editor, thanks for generous comment, actually, we have employed simple linear models to test hypothesis one, two, and three. In addition, we have employed multiple linear model with force entry technique based on expert biostatistician's opinion to test hypothesis four. Based on that, no logistic regression was utilized.*

▪ Within the causal inference framework, the authors can establish the acyclic directed graph; but current analysis is not methodologically correct. for instance, work environment is a confounder based on figure 3; so the independent effect of work environment on intention to quit should be -0.226; however, there should be many other confounders, which the authors did not fully accounted for. thus, the independent effect of -0.226 is not valid.

*Dear editor, thanks for generous comment, actually, the independent effect of work environment on intention to quit was -.187-, not as you mentioned -0.226. In addition, when we added the mediator variable to the model, the b-value of work environment perception became smaller (from -.187 to -.096) which gives evidence to the mediation effect of communication satisfaction. These results were confirmed by an expert statistician. THANKS.*

**Validity of the findings**

▪ The current finding cannot be supported by the analysis/study design.

*Agreed, the study's design was corrected appropriately in Line 29 and Line 149.*

**Comments for the Author**

▪ The abstract cannot express the main idea of the study; especially, the result section is poorly written without any useful information; you need data to support your conclusion in the abstract to make it stand alone.

*Agreed, we have revised it and presented it in consistent and strong, see Abstract.*

# Reviewer 3 (Anonymous)

**Basic reporting**

- For a research article, there is no need for a separate section for "Literature Review" (page 9). Please incorporate this as part of the Introduction section.

    *Dear editor, we have revised it and incorporated in one section.*

- Please distinguish clearly between theoretical framework and conceptual framework. You mentioned theoretical framework on page 8, along with Figure 1 and Figure 2. However, your Figure 2 appears to be the conceptual framework of the study.

    *Agreed, we have corrected the information regarding framework in Line 134.*

- Table 3 is confusing. You reported results of regression models. I can only see independent variable in the model(s). Where is the dependent or criterion variable?

    *Agreed, we have revised all tables and made them separated with essential information to prevent confusion, see Table 3, 4, 5, and 6.*

- Why did you conduct MLR with only 1 independent variable and 1 dependent variable?

    *Dear editor, no conducted MLR with one DV and one IV. Actually, we used one MLR analysis with two predictors. See Table 6.*

- In MLR, researcher uses several independent/predictor variables to predict the outcome of a dependent/criterion variable. This is not reflected in this study.

    *Dear editor, this good comment was employed in this study. Particularly, in Table 6.*

- What is the regression equation?

    *Agreed, we have provided the simple and MLR equations accordingly in data analysis section and result section.*

**Experimental design**

- The Methods section needs more elaboration, in particular, data analysis.

    *Agreed, we have put more elaboration with cooperation of an expert biostatistician in Line 218-240.*

- It is inadequate to just mention multiple regression analysis. For example, you must clearly identify the independent/predictor variables (IV) as well as the dependent/criterion variables (DV) used in the MLR. Did you conduct a step-wise/forward/backwards regression? etc.

    *Agreed, we have put more elaboration with cooperation of an expert biostatistician regarding MLR variables and technique in Line 218-240.*

- I am also unclear why you brought in Spearman's Rank correlation. You can get a correlation matrix as part of the findings when you conduct MLR.

  *Dear editor, we have brought Pearson's correlation matrix to ensure the absence of multicollinearity among predictors.*

- Furthermore, to conduct MLR, both the IV and DV must be continuous variables. Spearman's Rank correlation is for ordinal data.

  *Agreed, it was typo, actually we have used Pearson' Correlation test as we have continuous variables. It was corrected in Line 222.*

- Looking at your hypotheses and results, I would think you wanted to do a prediction study. Please make sure that the statistical tests used is correct and clearly described in data analysis.

  *Agreed, the study's design was corrected in Line 149 and supported by the appropriate statistical tests in Line 218-240.*

**Validity of the findings**

- The authors used three study instruments in this study, However, reliability and validity of the findings were not reported.

  *Agreed, we have provided the required information in instrumentation section.*

**Comments for the Author**

- Please improve on the Abstract. It does not accurately reflect the content of the manuscript, in particular, Results.

  *Agreed, we have improved it and made it stand alone.*

- There is a need to distinguish between theoretical framework and conceptual framework and use them appropriately.

  *Agreed, we have corrected the information regarding framework in Line 114.*

- Methodology of the study needs more elaboration, in particular, data analysis.

  *Agreed, we have put more elaboration with cooperation of an expert biostatistician in Line 218-240.*

- Results of the study need to be clearly reported. Please recheck Table 3.

  *Agreed, we have revised all tables and made them separated with essential information to prevent confusion, see Table 3, 4, 5, and 6.*

---

## Round 0.3 · accepted · Accept

The authors addressed the reviewers' concerns and substantially improved the content of MS. So, based on my own assessment as an academic editor, no further revisions are required and the MS can be accepted in its current form.

A PeerJ staff Editor with expertise in this field has suggested you make the following edits during the proofing stage:

1) all assertions of novelty should be toned down by adding "to our knowledge"

2) all conclusions should be tempered by adding "our results suggest" or "our findings suggest", etc.